



# Field-validated imaging of decadal and seasonal changes in permafrost bedrock using quantitative electrical resistivity tomography (Zugspitze, Germany/Austria)

Riccardo Scandroglio[1], Samuel Weber[2,3], Jonas K. Limbrock[4], and Michael Krautblatter[1]

[1]Technical University of Munich, Chair of Landslide Research, Munich, Germany
[2]Climate Change, Extremes and Natural Hazards in Alpine Regions Research Center CERC, Switzerland
[3]WSL Institute for Snow and Avalanche Research SLF, Davos, Switzerland
[4]University of Bonn, Institute of Geosciences, Geophysics Section, Bonn, Germany

**Correspondence:** Riccardo Scandroglio (r.scandroglio@tum.de)



**Abstract.** Ongoing permafrost degradation in alpine regions requires monitoring methods that accurately decipher spatial and temporal dynamics. Electrical resistivity tomography (ERT) is widely applied in bedrock permafrost, yet its outputs are often interpreted only qualitatively. Quantitative evaluation of ERT results, however, is crucial for improving process understanding and enhancing predictions of permafrost-related slope instability. In this study, we present a 17-year monitoring of permafrost

5  rock slopes on Mount Zugspitze (Germany/Austria) with monthly ERT campaigns. ERT data are combined with rock temperatures at four depths to establish field-based temperature–resistivity calibrations and to validate existing laboratory-derived relations. Both approaches agree well in the freezing range; however, field calibrations tend to yield higher resistivities at subzero temperatures and reveal substantial spatial heterogeneity. Incorporating reciprocal measurements refines the existing error model, increases image resolution, and improves the identification of subsurface features. Over ten years, the measured

10  rock temperature increased by $1°$C, accompanied by a 25% decrease in resistivity. The permanently frozen surface decreased by 40%, with degradation rated up to $-4.2$ kΩmy$^{-1}$. Extrapolating these trends would result in the loss of 65% of permafrost within a decade. Thermal forcing controls the degradation; however, the observed conditions and projected increases in heat-waves suggest that newly unfrozen and connected fracture networks will enhance advective heat transfer. This is expected to accelerate permafrost thawing, thereby increasing the risk of slope instability. With these results, we demonstrate that ERT

15  monitoring can yield high-quality quantitative insights into long-term permafrost evolution and effectively track bedrock permafrost degradation across both decadal and seasonal timescales.



## 1 Introduction

In the last few decades, air temperatures have reached unprecedented record values during repeated summer heatwaves, with consequences also underground. Warming of frozen ground is well documented at all latitudes and altitudes worldwide (Biskaborn et al., 2019; Smith et al., 2022; Gruber et al., 2017; Masiokas et al., 2020; Aalto et al., 2018) with European mountain permafrost warming about 0.4°C in the last 20 years (Etzelmüller et al., 2020; Noetzli et al., 2024). This degradation is projected to continue in the future in response to climate change, but its magnitude and timing are uncertain, as the thermal response appears to be site-specific (Smith et al., 2022). In fact, it depends on material, topography, surface characteristics, slope histories, permafrost type (warm vs. cold), and, most specifically, ice content (Magnin et al., 2017; Haberkorn et al., 2021; Deline et al., 2021). Ice-poor bedrock permafrost, close to 0°C, has recently experienced the strongest warming (Magnin et al., 2023). However, permafrost degradation near the freezing point appears to be non-linear, due to latent heat effects (Hauck and Hilbich, 2024). Degrading permafrost has demonstrably led to increased slope instability (Krautblatter et al., 2013; Mamot et al., 2021), as evidenced by recent events such as rockfalls, rock glacier acceleration, debris flows, and other cascading phenomena (Huggel et al., 2012; Phillips et al., 2017; Walter et al., 2020; Deline et al., 2021). Therefore, it needs to be closely monitored.

Geophysics can support these investigations with deep and spatially extensive insights (Vonder Muehll et al., 2001). Electrical Resistivity Tomography (ERT) clearly detects permafrost due to the large resistivity contrasts between frozen and unfrozen conditions (Tsytovlch and Sumgin, 1937; Muller, 1947; Parkhomenko, 1967). Focusing on mountain environments, ERT was introduced in the early 2000s (Vonder Muehll et al., 2000; Hauck, 2001). It quickly evolved into 2D surveys (Hauck and Vonder Muehll, 2003; Kneisel, 2004; Marescot et al., 2003) with Krautblatter and Hauck (2007) first applying it to solid bedrock permafrost, followed by 3D surveys (e.g., Rödder and Kneisel, 2012; Duvillard et al., 2018; Scandroglio et al., 2021), and cross-borehole measurements (Phillips et al., 2023; Bast et al., 2024). Repetition of measurements, i.e., time-lapse ERT, proved its capability to monitor permafrost changes (Hilbich et al., 2008; Krautblatter et al., 2010; Pogliotti et al.; Scandroglio et al., 2021). Fundamental for reproducibility is a robust set-up (Kneisel et al., 2014), but only recently have standard procedures been suggested (Herring et al., 2023). The majority of time-lapse studies are composed of one single measurement per year at the end of summer for permafrost detection (e.g., Kellerer-Pirklbauer and Eulenstein, 2023; Pavoni et al., 2023; Cathala et al., 2024) and the repetition of surveys accounts only for 20% of the studies (Herring et al., 2023). Most of these were single repetitions (e.g., Magnin et al., 2015; Buckel et al., 2023), some monitoring occurred at irregular time intervals (Scandroglio et al., 2021; Etzelmüller et al., 2023; Noetzli and Pellet, 2024), and only a few sites were regularly monitored (Mollaret et al., 2019; Morard et al., 2024). Although knowledge of the minimum permafrost extension at the end of summer is sufficient for most sites, hazardous locations require a higher temporal resolution to interpret site-specific, complex hydrothermal dynamics at depth (Krautblatter et al., 2010; Kneisel et al., 2014; Offer et al., 2025). This enables the evaluation of rock instabilities (Keuschnig et al., 2015; Etzelmüller et al., 2022; Cathala et al., 2024) to inform the development of effective geotechnical solutions and appropriate risk-reduction strategies (Bommer et al., 2009; Deline et al., 2021). Autonomous time-lapse ERT can provide high temporal resolution (Kneisel et al., 2014; Doetsch et al., 2015). However, the analyses published so far are limited



to 2 years or have significant gaps due to frequent technical failures (Hilbich et al., 2011; Supper et al., 2014; Keuschnig et al., 2017). Therefore, autonomous ERT has rarely been used to examine long-term trends in permafrost change (Herring et al., 2023). The only known work presenting continuous decadal data with a monthly resolution is by Mollaret et al. (2019) at the Schilthorn (CH).

Quantification of permafrost changes is obtained by calculating the average resistivity of the whole model (Hilbich et al., 2008) or only of the area of interest (Kneisel et al., 2014). Quantitatively translating resistivity values into temperatures is a non-trivial task, as many factors influence the measurements (Krautblatter et al., 2010). Laboratory calibration of bedrock samples was first introduced by Krautblatter et al. (2010) and subsequently employed in several studies to enhance the interpretation of geophysical data in permafrost environments (e.g., Magnin et al., 2015; Etzelmüller et al., 2022; Scandroglio et al., 2021),

although without quantitative field validation. ERT validation is possible with borehole data (Vonder Muehll et al., 2000; Lewkowicz et al., 2011; Doetsch et al., 2015; Mollaret et al., 2019; Farzamian et al., 2020) to constrain the interpretation of the results. Notably, while many studies report both borehole temperature data and ERT measurements, a direct comparison and validation between the two is often lacking. Further, no known studies validated laboratory-based calibrations in the field.

Here, we present the first field-validated long-term temperature-resistivity monitoring over more than 10 years in bedrock

permafrost. This database originates from a high-alpine location (Mount Zugspitze, Germany/Austria, 2,800 m above sea level) and is unique due to its monthly resolution, the location of the electrodes in the permafrost core rather than on the surface, and the high-quality results achieved through manual measurements. Results are compared with those of Krautblatter et al. (2010) to validate the methods and investigate the effects of climate change.

This study aims to answer the following questions:

1. How reliable is laboratory calibration, and is it consistent with field calibration and observations?

2. Do enhanced error models improve the reconnaissance of thermal processes in bedrock?

3. What long-term trends and seasonal signals of permafrost degradation can be effectively captured with monthly resolution?

4. At the Zugspitze, can we quantify permafrost extent, its degradation over a decade, and anticipate its future development?

## 75  2   Material and Methods

### 2.1   Field site

Measurements took place in the Kammstollen, a private tunnel under the E-W oriented ridge of Mount Zugspitze (2962 m asl, Fig. 1a-b), located at the German-Austrian border. This mountain is part of the Northern Calcareous Alps, and its summit is composed of 600–800 m more or less uniform layers of Triassic Wetterstein back-reef fine-grained limestones, that are partly

dolomitised in the vicinity of the summit (Miller, 1962). Brecciated zones up to 1 m thick are present in the summit area (Ulrich and King, 1993), a relevant fault zone can be found in the investigation area up to the summit (Hornung and Haas, 2017), and karst dissolution on the Plateau is frequent and well documented (Wrobel, 1980; Wetzel, 2004). The study area is accessible all



year round thanks to the infrastructure and logistical support of the research station Scheefernerhaus (UFS), which is directly connected to the tunnel.

Long-term meteorological records by the German Meteorological Service (DWD) exist since 1900 on the summit (Fig. 1c). Air temperature increased drastically in the last 30 years, with mean temperatures in the last decade (2013-2022) reaching $-3.3°C$, which is $1.5°C$ warmer than the reference period 1961–1990. Additionally, the years 2011, 2020, 2022, and 2024 were the warmest years ever recorded. Mean annual precipitation is more than $2,500\,mm$ of which 80% is snowfall, but the steep north walls remain snow-free for most of the winter. The hydrological behavior of clefts at this location has been 90 thoroughly measured for many years and interpreted by Scandroglio et al. (2025).

## 2.2 State of the cryosphere at the Zugspitze

The first records of permafrost presence on the Zugspitze are linked to the construction of its infrastructure (AEG, 1931; Körner and Ulrich, 1965; Ulrich and King, 1993). In recent years, 3D modeling has confirmed the possible presence of permafrost on the shaded and steep north side of the SW-NE ridge, which includes the site of this study, starting from approximately 95 2350 m above sea level, but suggests a rather warm permafrost (Noetzli et al., 2006). On the basis of this modeling, one N-S borehole was drilled in 2007 by the Bavarian Environmental Agency (LfU) about 30 m under the summit to document permafrost presence and evolution under the summit cable car, at a distance of 700 m from our field site (Gallemann et al., 2017, 2021; Wagner et al., 2023). These measurements support the modeling, showing average core temperatures between $-1.2$ and $-0.7°C$, with a warming trend of 0.4 K in the last ten years. Modeling by Gallemann et al. (2017) forecasts the 100 disappearance of this lens by 2070.

Krautblatter et al. (2010) conducted the first ERT monitoring in the tunnel in 2007, setting the basis for this study. The tunnel is composed of a main tunnel (MT) and a side tunnel (ST). Ice accumulated on the whole ground of the side tunnel and in some parts of the main tunnel, reaching a depth of up to 50 cm in some locations. Recently Mamot et al. (2018, 2021) conducted intense geophysical measurements and mechanical modeling to monitor an ice-controlled instability on the ridge 400 m from 105 the tunnel. This location showed strongly degrading permafrost due to the thermal influence of the southern slope on a thin ridge. Other geophysical measurements in the area utilized passive seismic methods to document permafrost degradation (Lindner et al., 2021) and relative gravimetry to detect mass changes associated with hydrology (Voigt et al., 2021). Finally, Mayer et al. (2021) documented the dramatic recession of the three glaciers at the Zugspitze, which have been decreasing in size since the 1980s and could completely disappear within the next decade.

## 110 2.3 Monitoring setup

Bedrock electrical resistivity ($\rho$): Following a feasibility study conducted in 2007 and 2008 (Krautblatter et al., 2010), monthly monitoring commenced in 2014. The setup was based on the feasibility study and remained constant to obtain comparable measurements. Measures were undertaken along 6 transects with electrode distances of 4.6 m and 1.5 m using Wenner and Schlumberger arrays. Problematic electrodes with poor coupling, or ice cover, have been consistently replaced. The monitoring





**Figure 1. Study site overview.** a) Zugspitze summit, with the DWD meteorological station and the LfU borehole. In red, the main tunnel, going from the research station (UFS) at 2650 m asl (south) to the exit in the north slope at 2800 m asl. ©OpenStreetMap contributors 2023. Open Data Commons Open Database License v1. b) Zoom of the monitoring site showing the main and the side tunnel, the ERT electrodes, the temperature loggers, and the two temperature transects (A-A' and B-B'). In the background, one exemplary tomogram from Krautblatter et al. (2010) shows the permafrost lens. A red line shows the approximate location of the fault zone. c) Mean annual air temperature (MAAT) at the DWD station. The thick red line shows the 30-year backward moving mean, while the black lines show the average for the selected period. d) Data availability for temperature logger (upper graph) and ERT (lower graph).





settings are presented in the supplementary material (Fig. S1). Between 2017 and 2018, data availability and quality were limited due to repeated cable failures; therefore, 14 measurements from this period have been excluded from this analysis.

Rock temperature ($T$): Hourly monitoring follows Krautblatter et al. (2010) with updates and extensions as listed in Table 1 and hereafter. i) The old loggers Type UTL-1 were substituted with new loggers Type Geoprecision with higher accuracy and reliability. ii) New loggers were installed along the main tunnel at 40 cm depth for rock and air temperature (MT and AT). iii)

One logger has been installed outside to measure rock surface temperature (RST) at 15 cm depth. The analysis presented here focuses on the period after 2018, where complete data have been recorded. Data availability is presented in Fig. 1d.

**Table 1.** List of all temperature loggers. MT = Main Tunnel, ST = Side Tunnel

| Transect | Name | Nr. | Type (Precision) | Location | Distance | Time | Note |
|----------|------|-----|------------------|----------|----------|------|------|
| A-A' | ST | 4 | Geoprecision M-Log 5W ($\pm 0.1^\circ$ C) | Side tunnel (ST), rock | 5 m | 2019-2024 | substitute UTL |
| B-B' | MT | 20 | iButton DS1922L ($\pm 0.5^\circ$ C) | Main tunnel (MT), rock | 4.6 m | 2020-2024 | new |
| B-B' | AT | 3 | Geoprecision M-Log 5W ($\pm 0.1^\circ$ C) | Main tunnel (MT), air | 46 m | 2020-2024 | new |
| - | RST | 1 | Geoprecision M-Log 5W ($\pm 0.1^\circ$ C) | outside, rock | - | 2022-2024 | new |

## 2.4 Data analysis

We linearly interpolated single temperature measurements along the tunnel from 2019 to 2023, as if they belonged to a unique "virtual" borehole. Values from the side tunnel (Transect A-A') were used for the field calibration together with values of

$\rho$ extracted from raw data, as normally done in laboratory calibrations. For validation, we analyzed the probability density function (pdf) of the apparent resistivity (Fig. 5), prior to inversion. Logger ST-5, located at a depth of 5 m from the surface, was excluded from the calibration because only summer resistivity measurements could be taken at this location, resulting in an insufficient amount of data.

Resistivity data were primarily collected using an ABEM Terrameter LS, with alternating use of an ABEM SAS 1000

and SAS 300C prior to 2018. Each measuring database was automatically imported into MATLAB to obtain the complete information. Data were filtered using the variance of resistance, i.e., the standard deviation divided by the mean, as computed by the measuring instrument, to remove systematic errors. For the ERT inversion, we utilized an updated version of the software *CRTomo* (Kemna et al., 2000), which features significantly shorter computing times.

Further on, the error model for the inversion was updated with new measurements. Image quality is highly dependent on the

error estimates for the inversion: an overestimation smooths the picture, reducing the resolution, while an underestimation leads to artifacts (Binley et al., 1995; LaBrecque et al., 1996; Slater et al., 2000; Lesparre et al., 2017; Tso et al., 2017). The random error ($e$) can be quantified using the difference between normal ($R_N$) and reciprocal ($R_R$) resistivity measurements, although this is only a measure of precision and not accuracy (LaBrecque et al., 1996) and does not represent the total experimental error. The error is a linear function of the mean resistance $R$ (Eq. 1), depending on an absolute ($a$) and a relative ($b$) term.

$$|R_N - R_R| = e = a + bR \tag{1}$$





To solve this, Slater et al. (2000) expresses $R$ as a function of $e$, removes obvious outliers ($e > 10\% \, R$), and obtains $a$ and $b$ from the envelope of all remaining measurements. Koestel et al. (2008) instead rejected only values of $e > 100\% \, R$ and then divided the measurements into logarithmically equally sized bins. The standard deviations of the reciprocal errors in each bin were least-square fitted with the linear error model (Eq. 1). Lesparre et al. (2017) adapted the approach of Slater et al. (2000) to

fit the discrepancies between readings acquired at different times, proposing data error estimates for time-lapse measurements.

To accurately analyze temporal changes, it is first necessary to correctly classify the different areas of the tomogram, such as the active layer and the permafrost core. Some authors manually define an area of interest (Kneisel et al., 2014), while others choose an automated algorithm (Watlet et al., 2023). Here, the inversion results were clustered using the MATLAB $k$-means algorithm. The function analyzed all tomograms from 2014 to 2023 and computed for each cell of the grid the

squared Euclidean distance from cluster centroids, which is the mean of the cluster. Cells were then categorized into the nearest centroid. The number of clusters $k$ is defined by the user, and assignment is mutually exclusive. The clustering is not unique, since multiple solutions are possible.

## 3 Results

### 3.1 Characterization of thermal regime from temperature

The temperature distribution in the tunnels indicates permafrost presence in transect A-A' (Fig. 2). Due to repeated measurement gaps prior to 2019, it is not possible to determine longer temperature trends. Over the last five years, the increase in active layer thickness (ALT) from 8.5 m to 10 m, combined with the overall rise in annual maximum temperatures, indicates degradation of permafrost. The 10 m logger has now reached $0°$C and is likely to overcome this threshold in the next years. No permafrost is recorded in transect B-B' (Fig. S2 in the supplementary material), only seasonally frozen rock between electrodes

E30 and E39. All loggers in the tunnel recorded their maximum in 2023, with a more pronounced increase in the area to the right of the side tunnel, specifically electrodes E32 to E41.

Outside the tunnel, the mean annual rock surface temperature (MARST) of the steep bedrock facing north is $-1.7°$C (Fig. S3c and f in the supplementary material). The correlation coefficient between the rock temperature inside the tunnel and the air temperature outside ranges from 0.77 for the side tunnel logger ST-5 to 0.4 for the ST-15 and ST-20 loggers

(Fig. S3a). As the logger depth increases, the time lag also grows, reaching up to 75 days, which represents the duration needed for the thermal signal to propagate to the permafrost core in the bedrock. The correlation between air temperatures inside and outside the tunnel is strong at electrode 30 (x = 143 m), near one of the tunnel exits, but becomes absent at deeper loggers (Fig. S3b), indicating the influence of additional processes at greater depths (e.g., solar radiation or snow cover from the south side slope).

A comparison of ST-Loggers with the LfU-borehole installed on the summit under the cable car (Gallemann et al., 2017) shows that ST-5 and ST-10 are quite similar to the LfU-borehole measurements in summer, while winter temperatures are much lower in the LfU-borehole (Fig. S4 and S5 in the SM). The site tunnel logger ST-5 has temperatures similar to those





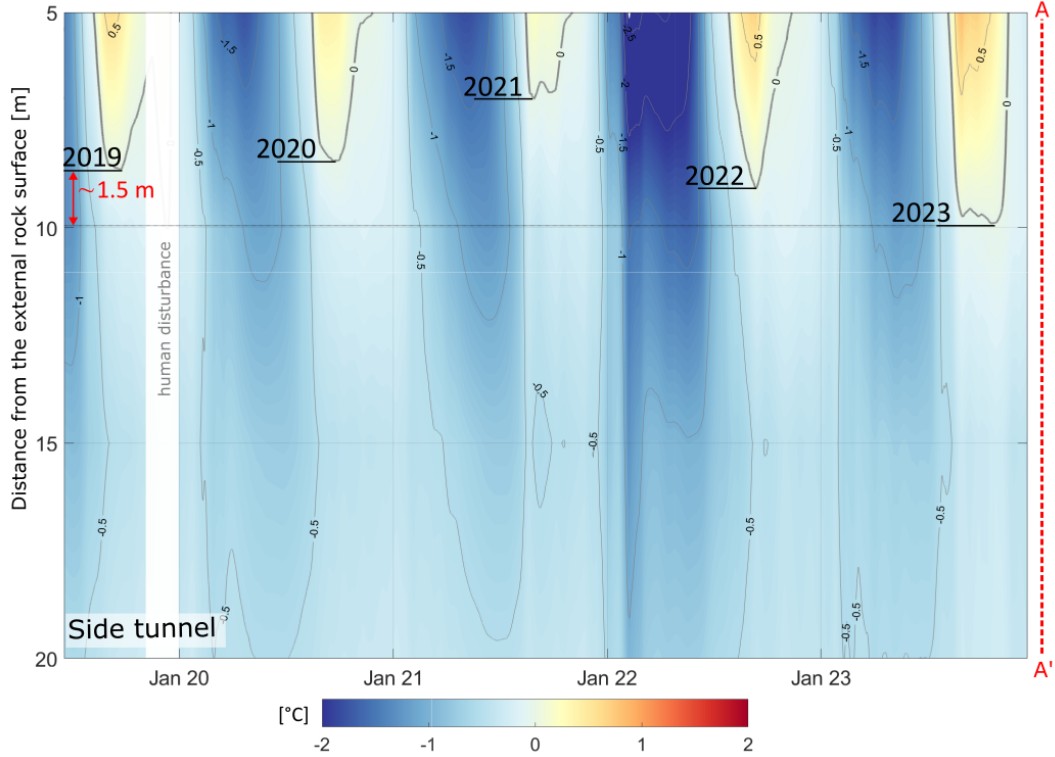

**Figure 2. Temperature profile in the side tunnel.** The location of the transect is visible in Fig. 1b (profile A-A').

of the borehole logger at a depth of 40 m. Deeper loggers (ST-15 and ST-20) are not reaching temperatures as low as in the LfU-borehole, even if the thickness of the ridge is comparable at the two locations.

Monthly visual mapping of ice and water presence in the main tunnel (Fig. S6 of supplementary materials) shows that ice extension on the ground reduced between 2019 and 2023, especially between electrodes E31 and E38, which is the area of the permafrost lens. This fits well with the temperature increase recorded by the MT-loggers. Water presence in the southern side of the tunnel/ridge is well documented from April to October by Scandroglio et al. (2025). Large quantities of water from snowmelt are also recorded in June and July in the northern part of the tunnel (electrodes E1 to E20, x = 200-275 m). Water

accumulation on the ground is common at the end of summer in the side tunnel, especially between the ST-5 and the north face. This water results from the melting of surface ice on the tunnel walls and ice-filled fractures within the tunnel.

**Field calibration of the $T - \rho$ relation**

The newly collected thermal information, together with the ERT measurements, allows us to validate the temperature-resistivity ($T$-$\rho$) laboratory calibration by Krautblatter et al. (2010), which was often used to interpret ERT field measurements so far (e.g.,





Magnin et al., 2015; Etzelmüller et al., 2022; Scandroglio et al., 2021). Figure 3 shows that, despite big differences in resistivity ranges, temperatures and resistivities have a high correlation.

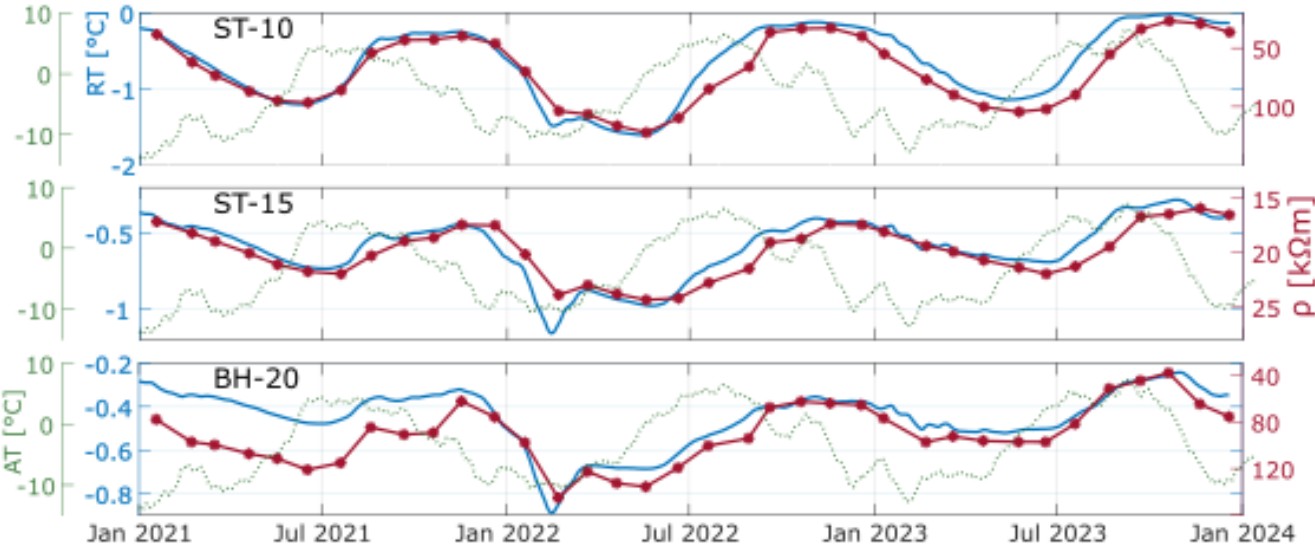

**Figure 3. Temperature versus resistivity.** Rock temperatures from the ST loggers in blue, and air temperatures from the DWD station in dotted green. In red, resistivity variations are interpolated in time from the monthly ERT measurement, with reverse axis direction for simplifying comparison. Other periods are available in Fig. S7 of the supplementary materials.

The differences in resistivity range lead to a strong spreading of data when plotting all the values together (Fig. 4a), but when analyzing each temperature logger separately, the correlation appears clear (Fig. 4b and Fig. S8b-e). This suggests that the relation between resistivity and temperature can not be assumed to be unique for this transect. Logger ST-5 presents only

a few values below 0°C: in the cold months, a thick ice cover in the tunnel limits the electrode connection, while in the warm months, positive temperatures are measured. Therefore, ST-5 will be excluded from our analysis. The other loggers present a good linear fit: the best is at ST-10 ($R^2 = 0.84$), while at ST-20, the fit is poor ($R^2 = 0.41$). Resistivities decrease with time, possibly due to a decrease in ice content; therefore, a yearly analysis could deliver a better fit. Generally, most of the recorded data are close to the freezing point, which should be taken into consideration. Supercooling effects and similar phenomena

from the calibration are not detectable in the field calibration due to the absence of freeze-thaw cycles in the permafrost core.

The freezing point at 29 kΩm agrees with values from the laboratory, but in the field, resistivity increases at much stronger rates than laboratory experiments can reproduce. Field calibration results for ST-10 and ST-20 can be resumed in the "average" Equation 2.

$$\rho = (-106 \pm 49)\,T + 29 \tag{2}$$

ST-15 shows much lower resistivity values than the other locations and behaves similarly to the laboratory calibration. This could depend on the degree of fracturing at this location.



**Figure 4. $T$-$\rho$ field calibration.** a) All values from 2021 to 2024. b) Calibration for ST-10, using only the values from the new T-loggers, from 2019. In red is the interpolating line, in green is the lab calibration from Krautblatter et al. (2010). c) Calibration lines for all loggers. In black, the average of logger ST-5, ST-10, and ST-20.

**Field validation of the $T - \rho$ relation**

To explain the big discrepancies in the field calibration, we analyze the rock that composes the tunnel. While the majority is made of compact rock (material A in Fig. 5), short parts of the tunnel are highly fractured due to the fault zone (material B in Fig. 5). As suggested by Scandroglio et al. (2021), we compare the probability density function (PDF) of the apparent resistivity of all the ERT measurements computed in the permafrost area (in the side tunnel). The ERT transect on the right (Fig. 5d) presents only one peak, while on the left (Fig. 5c) two peaks are present. Supposing that each peak relates to a different rock type, this indicates the presence of two materials in the left area, as confirmed by field mapping (Fig. 5a). Material A has an







**Figure 5. Comparison of ERT raw data for the two resistivity transects in the permafrost area.** a) Photos from the tunnel. Material A, at electrode E36, is pure bedrock. Material B, at electrode E49, is mostly fine-grained material. A mix of materials A and B with fractures of variable width is also possible. b) Areas of interest of the ERT transects in the side tunnel: violet for the left and yellow for the right transect. c-d) Probability density functions of apparent resistivity of the whole transect for the warmest months. Figure c) shows two peaks, indicating the presence of two main signatures, while Fig. d) presents only one. The year 2023 recorded the lowest apparent resistivities in both cases.

average apparent resistivity of 50 kΩm, while material B presents average values of 15 kΩm. A combination of both materials
would likely produce results similar to material B, as current follows the easiest path of lower resistivity; however, this should
be further investigated. PDF-curves for material A on the right side of the tunnel clearly show variations with time towards
lower values, while smaller changes are recorded on the left for material A and no changes for material B.





**Figure 6. Error model differences.** a) Error model according to Slater et al. (2000), each dot represents a reciprocal measurement. b) Relative differences between inversions with the old and the new error model for 2022. Resistivity values in $\log_{10}(\rho)$. Numbers express the overall change as a percentage. c) Zoom: Comparison of the permafrost extension in July. The upper figure is obtained using the old error model, and the middle figure is obtained using the new model. The lower image is a zoom of b).





## 3.2 Error model updating

During previous research at this site, Krautblatter et al. (2010) analysed 3000 reciprocal measurements according to Koestel
et al. (2008) and obtained values of $a = 48\ \Omega$ and $b = 0.08$. We recorded reciprocal measurements for four months in summer
and autumn to validate these results and analyzed them using three methods, as described in Slater et al. (2000); Koestel et al.
(2008); Lesparre et al. (2017). All methods suggest that the previous absolute error ($a$) was strongly conservative. The newly
computed value of $a$ approaches zero ($a = 0.25\ \Omega$), as Fig. 6 shows. The value of $b$ by Krautblatter et al. (2010) appears to be
correctly estimated according to Slater et al. (2000), but could assume lower values ($b = 4\ \%$) according to Koestel et al. (2008)
and Lesparre et al. (2017). For the following analysis, we will use the error estimation from Slater et al. (2000), as lower values
of $b$ can lead to overfitting in the tomograms.

In Figure 6b, we present the inversion differences between the error model chosen for this study and the model previously
used in Krautblatter et al. (2010), applied to the inversion of the ERT monitoring data from 2022. Using the updated error model,
resistivities in the permafrost core (x = 65 to 115 m) are higher in both winter and summer. The maximum differences occur
from May to July, especially on the right of the side tunnel (an increase > 10 %), while the smallest differences are between
January and April. An exception is represented by the area between y = 0 and −10 m right of the side tunnel (Fig. 6c), which is
very close to the rock surface. Here, resistivities are generally lower, with peaks during the warm months: this better represents
the active layer dynamics measured with temperature loggers. Strong differences are also evident in the area corresponding to
the fault zone crossing the permafrost body. The inversion with the updated error model reports lower resistivities along the
fault axis; however, it appears that these values do not extend to the main tunnel, as field observations would suggest. The area
between x = 150 and 200 m shows higher resistivities, but only in the winter months. The tunnel in this area is covered by
concrete (indicating a more fractured bedrock), and it is seasonally covered by ice on the ground and on the walls. Furthermore,
smaller differences are present at the tunnel surface (y < −25 m): here, frozen patches can be found at different x-positions,
but the most evident is at x = 135 m, which corresponds to one exit in the tunnel where snow normally accumulates in winter.
Lower resistivities are produced by the inversion at x > 225 m from January to July, with new values reaching −10 %.

## 3.3 Characterization of thermal regime from resistivity

**Long-term signal**

Figure 7a shows all ERT inversion results from 2007 to 2023 using the updated error model. This allows the detection of
features, listed hereafter, that are confirmed by field observation (Fig. 8a and Fig. S10 of the supplementary materials). a) The
active layer is visible from 2014 on, at the right side of the tunnel next to the permafrost core (x = 100 and 150 m). This
feature follows the structure of the external slope and reaches minimum values from July to September. b) From December
to April, a temporary frozen part appears at the center of the image (x = 150 to 200 m). This is an area of the tunnel with
reduced rock thickness and a steep bedrock slope outside. c) The constantly unfrozen part between 200 and 250 m fits with a
less steep debris-covered area outside the tunnel, which is snow-covered in winter. d) Some frozen patches can be found along
the tunnel (y = −25 m, lower border of each image): while they are present all year long in 2014 and 2016, these features







**Figure 7. ERT inversion results.** Tomograms from 2007 to 2023. A reference tomogram with measures can be found in the following figure.




strongly reduce, up to disappearing, from 2018 onward. e) The fault zone crossing the permafrost is evident in all tomograms. This area exhibits lower resistivities compared to the surrounding area, without a clear thermal explanation. f) The constantly lower values at the beginning of the tunnel represent the area with the biggest distance to the north slope. In addition to these features, measurement errors are also evident, thanks to the new error model, for example, in November 2020.

Long-term trends clearly present a reduction of the frozen areas, as confirmed by Fig. 8b. While frozen areas increased contradictorily between 2007 and 2014 from 31% to 40% of the total area, they drastically reduced from 2014 to 2020 down to 25%. This corresponds to a 40% reduction compared to 2014. The last four years present little variation in the minimum frozen area. These trends are also visible in the winter months, but are less pronounced.

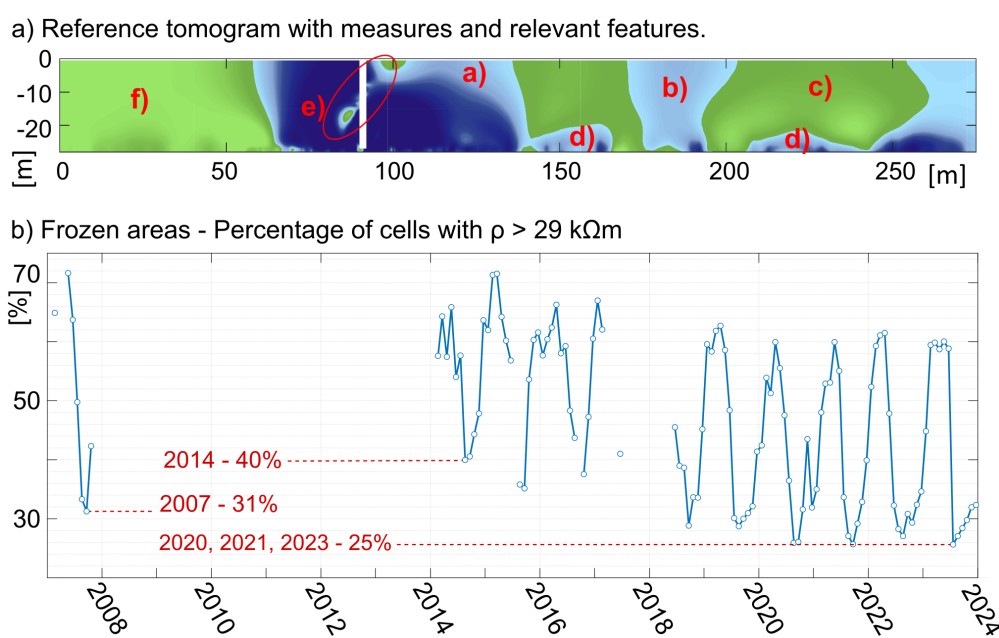

**Figure 8.** a) Reference tomogram with measures and highlighted features, same color bar as the previous image. b) Long-term development of the frozen areas, from 2007 to 2023.

**Image clustering of the tomograms with *K*-means**

The best-fit between measurements and the previously analyzed features is obtained for a clustering with $k = 5$. The results are presented in Figure 9. Class 1 shows the highest $\rho$ values, which correspond to constantly frozen rock. This class has been remarkably consistent over the last 10 years and represents the core of the permafrost. Classes 2 and 3 include cells that are, respectively, mostly frozen and partially frozen, representing rapidly degrading permafrost and the active layer. Both classes are experiencing the strongest decrease in $\rho$ ($-4200$ and $-2330$ $\Omega$my$^{-1}$, Fig. 9b). Class 4 comprehends mostly unfrozen

cells with a slightly decreasing 10-year trend and represents feature c) of Figure 7. Class 5 encompasses the unfrozen area at the beginning of the tunnel, which has exhibited slightly increasing values over the past decade but has undergone constant



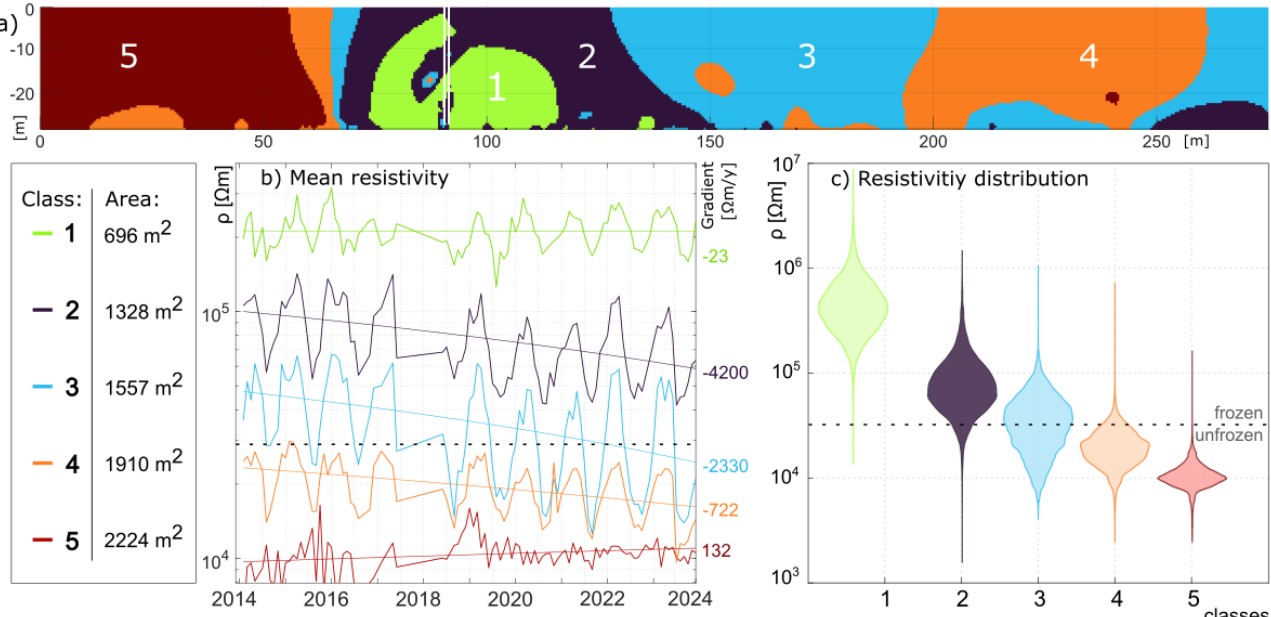

**Figure 9. *k*-means clustering.** a) Classified tomogram. b) Mean resistivities for each class on a logarithmic scale. On the left, the areas covered by each class are listed, and on the right, the corresponding gradients are shown. c) Violin plot of all resistivity values included in one class.

development recently. The distribution of values in Figure 9c shows all values and not only the mean. In particular, values in class 2 are mostly frozen, while those in class 3 are half frozen and half unfrozen.

**Quantification of changes in permafrost extent**

Knowing the grid size, we can estimate the size of the permanently frozen area. It covers circa 2,000 m², of which 700 m² is in class 1 (35%) and 1,300 m² is in class 2 (65%). The external morphology suggests a vertical extension of the lens of 20-30 m, which would result in 40,000-60,000 m³ of frozen rock. The remaining temporarily frozen and unfrozen areas (classes 3, 4, and 5) cover 5,700 m², about 75% of the tomogram. Given the strong decreasing rates of the outer layer (class 2), up to 39000 m³ of permafrost are very likely to suffer irreversible thermal degradation, potentially leading to its disappearance. By

linear extrapolation of the gradient computed in Fig. 9b, we expect this process to happen within the following decade. This assumption significantly simplifies the complex physical processes, which are not necessarily linear and therefore should be considered with some degree of uncertainty. Still, given the actual development of climate change, the trend is unquestionable.

**Quantification of seasonal variations**

For detecting seasonal trends, we utilized the mean $\rho$ value of the whole ERT tomogram. The maximum values are recorded in

April or May, the minimum around September (±1 month), as shown in Fig. 10a. In the summer months, there is a clear shift





in time towards lower values of $\rho$, while this is less evident in the winters, which present a higher variability. Resistivities in 2023 (dark red) indicate a prolonged winter with low values, a drastic drop in resistivities from June to July, and a long summer season with extremely low values from July to October. Comparing ten years of records reveals three distinct phases. The first is a gradual increase in resistivity from October to March/April; then there is a sudden decrease between May and July, and finally, a stable phase with little variation in resistivity from August to October.

Seasonal averages in Fig. 10b show that the resistivity decrease in the last ten years is more pronounced in the spring and summer months, reaching more than 0.9 kΩm per year. Winter values are decreasing at a rate of less than one-third. While winter and spring values have strong variations from year to year, summer averages follow a clear linear trend.

Figures 10c shows that the tomogram´s mean $\rho$ is strongly correlated with the mean monthly air temperature with an hysteresis pattern. Resistivity reacts with about a two-month delay to temperature changes, producing an elliptical path. The change in time (described by colors) shows the evolution over the last 10 years towards warmer air temperatures ($+2°$C in summer), which led to lower resistivity ($-8$ kΩm). 2023 recorded the lowest value both for $\rho$ and MAAT: $4.8°$C and 19 kΩm.

## 4  Discussion

We present 15 years of monthly ERT profiles and investigate permafrost dynamics in steep bedrock applying a newly developed temperature-resistivity ($T - \rho$) field calibration and an updated error model, compared to the first studies by Krautblatter et al. (2010). Newly available data, improvements in methods, and updated analyses have significantly enhanced the quantification of both short-term permafrost dynamics and long-term degradation.

### 4.1  Linking temperature and electrical resistivity: comparison between laboratory and field observations

Precise knowledge of the $T - \rho$ correlation is crucial for a correct interpretation of the thermal state of the permafrost core derived from resistivities. Many laboratory calibrations are nowadays available for different sites and materials (examples in Herring et al., 2023; Limbrock et al., 2025), but most studies do not validate them quantitatively in the field (e.g. Magnin et al., 2015; Duvillard et al., 2021). Only Mollaret et al. (2019) conducted field calibration before, but without comparing it with laboratory values. Offer et al. (2025) compared laboratory $T - \rho$ relations with field values, but without real calibration. Information at permafrost core depth is often missing, as no known studies have compared field and laboratory measurements.

Our new field calibration, compared with laboratory calibration by Krautblatter et al. (2013), presents similar values for the freezing point, but higher resistivity gradients at low temperatures. Similar results are obtained when comparing field calibration by Mollaret et al. (2019) with laboratory measurements from the literature (Hauck, 2001). In this case, the field results also exceeded the values obtained in the laboratory. Knowing that laboratory tests are typically conducted on intact rock samples, we can hypothesize that laboratory results provide a valid approximation for a saturated matrix. The higher resistivity values obtained in the field can be explained by the presence of ice in frozen discontinuities.

Both field and laboratory calibrations at our site agree on the freezing point around 29 kΩm; however, the interpretation of laboratory measurements above this point should be taken with caution. In fact, it is impossible to accurately reproduce



**Figure 10. Seasonal changes of mean resistivity.** Average value of the entire ERT tomogram, from 2014 to 2023. a) Seasonal variation for all years. b) Average values of 3 months, with trendline and gradient: February, March, and April in blue, May, June, and July in red, and August, September, and October in yellow. c) Comparison of the tomogram mean resistivity with the mean monthly air temperature, measured at the DWD station on the summit, both values as a 3-month moving mean.



the steadily variable alternation of bedrock, fractures, ice/water, and voids available in nature with a sample; however, this composition strongly influences ERT results. This is evident in the high variability of the calibration results presented here, despite our temperature loggers being located just 5 m apart from each other. For this reason, caution should be used when extrapolating these results for other locations, even at the same site. We suggest that field calibrations are totally valid only for the measured spot. Further limitations are a reduced temperature range in nature and the lack of freeze-thaw cycles when measuring permafrost. Recent studies confirm the presence of hysteresis effects in field measurements and demonstrate their importance in the energy balance of permafrost (Limbrock et al., 2025; Luo et al., 2024; Tomaškovičová and Ingeman-Nielsen, 2024).

Our measurements recorded resistivities greater than 200 kΩm, values that exceed the laboratory calibration range. These values have been explained by Krautblatter et al. (2010) in terms of ice presence, but not numerically proven. Our results indicate that laboratory calibration should be validated in the field and, if necessary, refined according to the field data.

### 4.2 Influence of the error model on long-term, quantitative ERT measurements

The chosen procedure, the criteria for outlier filtering, and the placement of bins can affect the quantification of the error to varying degrees (Koestel et al., 2008; Tso et al., 2017). Different arrays generate different types of errors, corresponding to distinct error models. Further complexity arises when multiple measurements are combined into a single dataset, as in this case. In addition, the reciprocal error may vary over time during long-term monitoring due to changing external conditions, such as seasonal fluctuations in water state or long-term variations in underground water content. Given these factors, the best inversion would theoretically require one error model for each array and each epoch, which would be extremely time-consuming. As a consequence, the use of reciprocal errors is very rare in field applications (Herring et al., 2023). When reciprocal measurements are conducted, most studies use a uniform error level for all configurations and time steps to reduce the degrees of freedom (Koestel et al., 2008; Lesparre et al., 2017; Tso et al., 2017), thereby ensuring consistency among ERT images at different times (Krautblatter et al., 2010; Maierhofer et al., 2024). With the new computation of the error model suggested here, i.e., the drastic reduction of the absolute term $a$, we possibly introduce some artifacts in the inversion results. Still, the significant advantage gained is a substantial increase in resolution, which enables us to investigate permafrost degradation with much greater detail.

### 4.3 Permafrost degradation according to ERT

New temperature calibrations and error models enable us to accurately interpret the inverted resistivities, particularly in the permafrost core. Results confirm the presence of warm permafrost close to the melting point, and its degradation is well quantifiable. Still, complex inter-annual behavior due to latent heat effects during thawing, seasonal water infiltration, and variable snow cover makes the processes highly non-linear (Hauck and Hilbich, 2024) and therefore difficult to interpret. The degree of fracturing also seems to contribute to this non-linearity. A survey of fractured zones was already presented in Krautblatter et al. (2010), but the effects of the fault zone were neglected. Here, thanks to the new field calibration, differences are distinctly highlighted. Clustering, as suggested by Delforge et al. (2021) and Watlet et al. (2023), also provided powerful insights in this direction. Thanks to simple algorithms, large amounts of data are used to detect similar patterns, yielding an



objective, data-driven analysis. This allows for a more robust interpretation of results than the "user-defined areas of interest", which are mostly rectangular, as previously used in the literature (Kneisel et al., 2014).

Despite this complexity at high temporal and spatial resolution, when looking at slope scale over one decade, it is evident that ERT values strongly correlate with rock and air temperatures, as shown in Figure 11b. Mean resistivity of the tomograms

decreases from 37 kΩm in 2018 to 28 kΩm in 2023 ($-25\%$). The decrease is stronger from 2018 to 2021, while in the last 3 years, it appears to be less pronounced. Meanwhile, air temperatures have steadily increased since 2012, rising from $-4.2°$C to $-3.2°$C.

This supports the theory that external thermal forcing and conductivity are the primary processes promoting permafrost degradation here. Contradicting trends, such as the increase in frozen areas between 2007 and 2014, as shown in Figure 8, can

also be explained by thermal forcing (e.g., an increase in air temperatures). In fact, the hydrological year 2006/2007 recorded extreme temperatures, $1°$C warmer than the 30-year average. On the contrary, from 2007 to 2014, temperatures were mostly below the 30-year average (Fig. 11b, inlet). Therefore, 2007 can be considered an anomaly, and the measurements in that year are not representative of the decade 2000-2010.

Analyzing the resistivity ratio between active layer and permafrost $\rho_{PL}/\rho_{AL}$ as suggested by Hauck and Hilbich (2024),

we compare clustering class 1 with class 2 and class 1 with class 3 and focus on the maximum values of each year (Fig. 11a). Although showing different variations, both ratios increased over the last 10 years. The ratio class 1/class 2 closely follows the trends of mean annual air temperature (MAAT): it decreased in 2018 and in 2021 after mild years, while it increased in all other years. In 9 out of the last 10 years, MAAT was above the long-term average of $-3.8°$C, with records reaching $-2.6°$C in 2019/2020 and 2022/2023. Excluding the possibility that the permafrost core has increased its resistivity ($\rho_{PL}$), this trend can

be explained only by a decrease in resistivity in the active layer ($\rho_{AL}$), to a stronger degree than the decrease in the core, as confirmed by Figure 9b. Compared with Figure 8, it is remarkable that, although the number of frozen cells has not decreased since 2021, the ratio $\rho_{PL}/\rho_{AL}$ increased in 2022 and 2023. This means that even if frozen areas do not diminish, resistivities reduce, indicating that non-linearly degrading processes are occurring in the active layer, and that sudden degradation might occur in the next few years. On the contrary, it seems that class 1/class 3 stabilized in the last four years, possibly indicating

that degradation already reached a stable level in these areas.

The resistivity ratio between the active layer and the permafrost in our data falls within the same range as reported by Hauck and Hilbich (2024). However, while they observed a long-term decrease in this ratio, our results show a contradictory overall increase. This suggests that, at our site, the degradation of the active layer has not yet significantly impacted the permafrost core. Notably, our unconventional measurement setup—conducting measurements from the inside out rather than from the surface

may influence this interpretation. In typical surface-based ERT surveys, accuracy decreases with depth due to the limited number of data points, resulting in higher confidence near the surface and lower confidence at greater depths. In contrast, our configuration yields the opposite: reduced accuracy in the shallow active layer and improved accuracy in the deeper permafrost core.





**Figure 11. External thermal forcing drives resistivity changes in the active layer.** a) Resistivity ration: class 1/class 2 in dark blue, class 1/class 3 in light blue. The red lines unite the minimum values for each year. Lower bars: mean annual air temperatures (MAAT) of hydrological years from 2014 to 2024. b) The increase in overall mean resistivity (in red) is well explained by the increase in air temperature (in black). c) Inlet: MAAT of hydrological years from 2001 to 2014.

## 4.4 Future development of permafrost degradation and slope stability

At this site, permafrost degradation is mainly driven by thermal forcing, as shown in Figure 11a. Future heat waves are projected to increase in magnitude and frequency (Lin et al., 2022). As a result, this trend is expected to persist in the coming decades,





particularly for classes 2 and 3, although the permafrost core (class 1) will also ultimately be impacted. The fault crossing the tomogram, clearly visible thanks to the newly developed error model, may further accelerate degradation due to its high permeability. In fact, when unfrozen, it could allow large volumes of water to infiltrate, as demonstrated by Scandroglio et al.

(2025), introducing advective heat transport processes that compound the thermal degradation. Consequently, atmospheric forcing is expected to play a significant role in the dynamics of permafrost degradation; however, its influence remains difficult to quantify.

On steep slopes, the increase in rock temperatures is linked to irreversible ice losses (Hauck and Hilbich, 2024). Although this is not expected to strongly influence the hydrological cycle in alpine environments, it has strong stability consequences for

slope stability (Krautblatter et al., 2013). This might also be true for the analyzed area, where the fractured zones might be a source of new instabilities, also considering the almost vertical dipping of this fracture area.

## 5    Conclusions

This study compiles 109 ERT measurements collected over 17 years at monthly frequency from the Kammstollen Tunnel, 2,800 m above sea level on Mount Zugspitze (Germany/Austria). Developing a field calibration for the temperature-electrical resis-

tivity relation, improving the error model of the inversion, and automatically clustering the results, advances the understanding and quantification of permafrost degradation in steep rock slopes.

- ERT field-based calibration with rock temperatures over a five-year period validates laboratory-based calibrations, which have become a standard approach in the literature. While both methods show good agreement on the freezing point, field data reveal significantly higher resistivity values at sub-zero temperatures and highly variable results. This highlights the

need for cautious interpretation of tomograms based solely on laboratory calibration.

- A correct estimation of measurement errors is essential for a correct inversion and interpretation of ERT data. Regular validation and update of the error model over time can significantly enhance the detection of thermal processes in long-term monitoring.

- Mean resistivity of the tomogram shows a decrease of 25% in the last 10 years, with more enhanced degradation in the

summer months, where the overall decrease rates reach $-0.9\ k\Omega my^{-1}$.

- Frozen cells decreased by almost 40% in the same period, and the active layer suffered the strongest losses, with rates between $-4.2$ and $-2.3\ k\Omega my^{-1}$.

- The strong meteorological variability between years requires at least monthly monitoring in the summer and autumn months to precisely assess long-term degradation. Monitoring the spring and winter months is crucial for a comprehen-

sive understanding of the processes and an accurate evaluation of the yearly trends.

- The extent of the permafrost lens in 2023 can be estimated in 2,000 m$^2$ (Class 1 and 2) - up to 60,000 m$^3$. The actual degradation rates indicate that approximately 1,300 m$^2$ (class 2) - up to 39,000 m$^3$ - will become unfrozen within the

next decade. Still, degradation is not always linear, and this estimation does not account for thermal advection caused by infiltrating water (Scandroglio et al., 2025), which could entirely enhance this phenomenon.

– The proposed resistivity-temperature monitoring allows a quantitative investigation of changes in the thermal regime of permafrost. Compared to previous studies, the results confirm the presence of the permafrost core, but indicate differences in its spatial extent, providing a more accurate representation of its dynamics.

These innovations enhance the ability to detect and predict permafrost degradation and the consequent bedrock instabilities with greater spatial and temporal precision. These findings help decision-makers assess the increasing risks that rising temperatures
pose to both society and infrastructure.

*Data availability.*  All data analyses and visualizations for this study have been conducted in MATLAB. Resistivity and temperature data are available at the following link: (will be provided before publication).

*Author contributions.*  RS designed the study, performed the field measurements, and conducted the data interpretation. JL supported the inversion of ERT data, and SW supported the development of the manuscript and data interpretation. MK designed, financed, and supervised
the study. RS prepared and revised the manuscript with final approval from all authors. SW and JL improved the draft.

*Competing interests.*  The authors declare that they have no conflict of interest.

*Acknowledgements.*  We extend our special thanks to the Environmental Research Station Schneefernerhaus and the Bayerische Zugspitzbahn Bergbahn AG for their logistical and financial support. We also thank our colleagues from the Chair of Landslide Research and the Chair of Engineering Geology for their continuous assistance and collaboration. Special thanks to all the students who helped in the field and provided
constructive ideas, especially Paul Schmid, Verena Soll, Saskia Brose, Frederik Reese, Leon Wassmann, Jonas Brixle, and Andrea Schmid. This research has been supported by the AlpSenseRely project, funded by the Bavarian State Ministry of the Environment and Consumer Protection (grant no. TUS01UFS-76976), and by the HydroPF project, funded by the TUM International Graduate School of Science and Engineering (IGSSE).



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
