# Peer review of "Field-validated imaging of decadal and seasonal changes in permafrost bedrock using quantitative electrical resistivity tomography (Zugspitze, Germany/Austria)"

_EGUsphere, 2025_

## Referee Comment (RC1)

Review of the manuscript entitled: "Field-validated imaging of decadal and seasonal changes in permafrost bedrock using quantitative electrical resistivity tomography (Zugspitze, Germany/Austria)"

I enjoyed reading this article. It is written in a clear and logical manner, presenting a great dataset and a compelling way of quantifying permafrost changes.

Overall, I believe the article is in good shape, with a few sections and figure that could benefit from more specifics. I have some suggestions and comments for the authors which I will list below.

Line 49: I think you mean automated.

Line 52: As above.

Line 62: Shallow borehole data has been used to calibrate automated ERT datasets, please see Cimpoiasu et al. (2024, 2025):

Cimpoiasu, M.O., Kuras, O., Harrison, H., Wilkinson, P.B., Meldrum, P., Chambers, J.E., Liljestrand, D., Oroza, C., Schmidt, S.K., Sommers, P., Irons, T.P. and Bradley, J.A. (2024), Characterization of a Deglaciated Sediment Chronosequence in the High Arctic Using Near-Surface Geoelectrical Monitoring Methods. Permafrost and Periglac Process, 35: 157-171. https://doi.org/10.1002/ppp.2220

Cimpoiasu, M. O., Kuras, O., Harrison, H., Wilkinson, P. B., Meldrum, P., Chambers, J. E., Liljestrand, D., Oroza, C., Schmidt, S. K., Sommers, P., Vimercati, L., Irons, T. P., Lyu, Z., Solon, A., & Bradley, J. A. (2025). High-resolution 4D electrical resistivity tomography and below-ground point sensor monitoring of High Arctic deglaciated sediments capture zero-curtain effects, freeze–thaw transitions, and mid-winter thawing. Cryosphere, 19(1), 401–421. https://doi.org/10.5194/tc-19-401-2025

Line 79: m of more or less

Line 95: "m asl" acronym was used above, please be consistent.

Figure 1:       b) attach color scale please; labels of main and side tunnels need to be slightly more readable

d) – caption: "temperature loggers"

d) Is that meant to represent data retention post filtering? it is not explained very well in text and caption. Please add more detail.

Line 132: I am assuming you are doing a timelapse inversion here, but you have not specified.

Line 172: "Side tunnel".

Line 196: The authors would agree that resistivity exhibits a different relationship with temperature above and below the freezing point, at which the value of 29 kΩm was measured. However, it seems like inferences about the freezing point are mostly reliant on a more thorough analysis discussed in another paper: Krautblatter et al. (2010). For clarity I

would encourage the authors to either a) explain in more detail the lab measurements they make references to or b) summarize the findings of Krautblatter et al. (2010).

Figure 5: delete space in "E 53"

b) I don't understand the color scheme

should be subplot or subfigure c) and d)

Line 209: I am not sure average resistivity a representative value for the rock/matrix type. Is this the average across all records?

Figure 6: "the lower image represents the subsection of b outlined by a black rectangle"

Figure 7: There is no X axis, so the text Line238-245 is very hard to follow.

Replace "following figure" with "figure 8".

3.3: You don't report any metrics regarding the %RMS of your inversions.

Section 3: With such a long record of measurements and having experienced some problems which lead to faulty datasets, isn't there some scope to add a section about "lessons learned" in reference to the monitoring setup. I am sure it will make an interesting read and addition to this work.

Figure 8: Is the max % coverage of frozen areas also changing?

There is no color scale for resistivity here.

Line 255: I think you need a justification for choosing that specific number of clusters.

Line 269: 39,000.

Figure 11: What is MMAT? What is mm12m/mm36m?

Section 4: You talk a bit about future scenarios in which a critical point of low permafrost might be reached, generating geohazards, such as rock/landslides. Given that you have an automated set-up and you have the potential to track the evolution of temperature and permafrost coverage in near real time, would it be scope to discuss some parameters one could be looking at in order to evaluate the state of emergency and perhaps form the basis of an early warning system?

Section 4: Are there any future plans for this site? One thing to consider would be to have an increased measurement frequency around the shoulder/warm season? (not sure if the battery power would be an issue). This way you can better capture the active layer evolution and heat transport.